# From integrability to chaos in quantum Liouvillians

Álvaro Rubio-García, Rafael A. Molina* and Jorge Dukelsky

Instituto de Estructura de la Materia, IEM-CSIC,
Serrano 123, Madrid, E-28006, Spain

* rafael.molina@csic.es

## Abstract

The dynamics of open quantum systems can be described by a Liouvillian, which in the Markovian approximation fulfills the Lindblad master equation. We present a family of integrable many-body Liouvillians based on Richardson-Gaudin models with a complex structure of the jump operators. Making use of this new region of integrability, we study the transition to chaos in terms of a two-parameter Liouvillian. The transition is characterized by the spectral statistics of the complex eigenvalues of the Liouvillian operators using the nearest neighbor spacing distribution and by the ratios between eigenvalue distances.

## 1 Introduction

Quantum chaos has gained a renewed interest in the last few years due to its relevance to important current topics of research like quantum thermalization, non-equilibrium dynamics of quantum many-body systems and quantum information [1–5]. In the standard Hermitian quantum mechanics valid for closed systems, the dynamical regime of the system can be characterized through spectral statistics [6]. Integrable systems present level clustering

and the nearest-neighbor spacing distribution follows the one-dimensional Poisson distribution $P(s) = e^{-s}$ [7], while chaotic systems present level repulsion with the $P(s)$ close to the Wigner surmise of Random Matrix Theory (RMT) depending on their symmetry class, $P(s) \propto s^{\beta}$ for small $s$, with $\beta = 1, 2, 4$ for orthogonal, unitary, and symplectic symmetries respectively, which is the content of the famous Bohigas-Giannoni-Schmit (BGS) conjecture [8]. The BGS conjecture is well founded now in the semiclassical theory, valid for systems with a proper classical limit [9–11] and supported by overwhelming numerical and experimental evidence in many different quantum systems [12–14]. The situation in many-body quantum systems is much less clear, although some theoretical advances have been recently made [15–17]. Due to the symmetry under exchange of fermionic or bosonic particles, the classical limit cannot be properly defined. Usually the BGS conjecture is assumed to hold also for many-body quantum systems, mainly based on numerical results, but a rigorous derivation is still lacking. The transition between the integrable and chaotic universal limits is non-universal, depending on the specifics of the particular system under study, although it has been explored in much detail for different systems [18, 19]. In the transition between the integrable and chaotic orthogonal cases, for example, some systems present fractional level repulsion with $P(s) \propto s^{\beta}$ with the value of $\beta$ varying continuously between the integrable case $\beta = 0$ and the corresponding RMT ensemble value $\beta = 1$, while others present full level repulsion but only for a fraction of levels [20]. Many systems, specially in the many-body case, show the former behavior. However, the semiclassical theory of Berry and Robnik for the transition predicts the latter one [19]. In this case $P(0) = F$, with $F$ given by the fraction of regular orbits in phase space of the classical limit for the model under consideration.

In open quantum systems the theory is much less developed, even if the first results came shortly after the proposal of the BGS conjecture [21]. Open quantum systems can be described by the Liouville equation, which characterizes the time evolution of the density matrix operator. In the Markovian approximation the Liouvillian is a linear non-Hermitian operator and the Liouville equation can be written as a Lindblad master equation [22]. The Liouvillian, then, has complex eigenvalues instead of the real energies of standard Hermitian quantum mechanics. The initial approach to the problem was to study integrable or chaotic Hamiltonians with a weak coupling to the environment. When the Hamiltonian was integrable, Grobe *et al.* studied the spectral statistics in the complex plane and found good agreement with the two-dimensional Poisson distribution [21]. In the chaotic limit there appears universal cubic repulsion $P(s) \propto s^3$ for small values of $s$ as in the Ginibre ensemble of non-Hermitian random matrices [23], although the details of the full $P(s)$ distribution depend on the symmetries of the non-Hermitian matrix [24, 25]. For an open quantum spin chain, the level spacing distribution in the transition from integrability to chaos has been fitted by a static two-dimensional Coulomb gas with harmonic confinement where level repulsion is given by the inverse temperature showing fractional level repulsion in the transition [26].

Very recently, the need for a different approach in the study of the integrable and chaotic properties of open quantum systems has been triggered as a consequence of the discovery of new families of integrable many-body Liouvillians [27–29]. Extending the class of exactly solvable and quantum integrable Liouvillians is an important step towards improving our understanding of open quantum many-body systems. The statistical properties of the complex spectra of random chaotic Liouvillians have been studied in a few recent works [30, 31]. However, the transition between the exactly solvable integrable limit and the chaotic limit in physical many-body Liouvillians remains mostly unexplored.

In this letter we will extend the model of Ref. [28] based on the SU(2) spin 1 Richardson model to a line of integrability within the rational Richardson-Gaudin (RG) class of integrable models. This new family of integrable Liouvillians has a rich and complex structure of the jump operators and allows for a simple parametrization along the line of integrability. We then

define in terms of a single parameter a Liouvillian that interpolates between integrability and a fully chaotic limit. With these model Liouvillians we characterize these transitions through the spectral statistics of the complex eigenvalues of the Liouvillian operators and by the average properties of the complex spacing ratios.

## 2 Integrable Richardson-Gaudin Liouvillians

We consider a system described by a Hamiltonian $H$ weakly coupled to a Markovian environment. The evolution of its density matrix can be expressed in terms of the Lindblad master equation [22]

$$\partial_t \rho = \mathcal{L}(\rho) = -i[H, \rho] + \sum_j \left( L_j \rho L_j^\dagger - \frac{1}{2} \rho L_j^\dagger L_j - \frac{1}{2} L_j^\dagger L_j \rho \right), \tag{1}$$

where $\mathcal{L}$ is the Liouvillian superoperator acting on the space of density matrices. The first term on the right hand side is the Hamiltonian commutator describing the unitary evolution and the second term takes into account the interaction with the environment through the Lindblad jump operators $L_j$. The time evolution of a Liouvillian eigenstate $\mathcal{L}(\rho_i) = \lambda_i \rho_i$ is given by $e^{\mathcal{L}t} \rho_i = e^{\lambda_i t} \rho_i$ and it can be shown that the real part of the eigenvalues is always non positive, $\mathrm{Re}\{\lambda_i\} \leq 0$ [22, 32, 33]. Furthermore, the Liouvillian spectrum consists of complex conjugate pairs $(\rho_i, \lambda_i)$, $(\rho_i^\dagger, \lambda_i^*)$, implying that a real eigenvalue corresponds to an hermitian density matrix. There is at least one state with zero Liouvillian eigenvalue, the *steady state* $\rho_{\mathrm{SS}}$, to which any initial state decays in the long time limit $\lim_{t \to \infty} e^{\mathcal{L}t} \rho = \rho_{\mathrm{SS}}$. As a consequence, the trace of any eigenstate with negative real part must be zero. The steady state can also be degenerate, in which case the system will decay to a linear combination of states within the degenerate manifold depending on the initial state.

Similar to Ref. [28], we study a chain of $L$ spins subject to a non uniform magnetic field $h_i$

$$H = \sum_{i=1}^{L} h_i S_i^z, \tag{2}$$

whose interaction with the environment is mediated by quantum jumps of the form

$$L_z^a = \sqrt{2\Gamma_0} \sum_{i=1}^{L} z_i^a S_i^z, \quad L_\pm = \sqrt{\Gamma_\pm} \sum_{i=1}^{L} x_i S_i^\pm, \tag{3}$$

where $a$ labels a set of different jumps and $z_i^a$ and $x_i$ are arbitrary jump amplitudes. As we will see, the equivalence of the gain $L_+$ and loss $L_-$ operator strengths, $\Gamma = \Gamma_+ = \Gamma_-$, is a necessary condition for quantum integrability. In the particular case of a single collective spin Hamiltonian this condition could be relaxed [29]. Note that for this choice the jumps become hermitian, $\left(L_z^a\right)^\dagger = L_z^a$, $(L_+)^\dagger = L_-$, which immediately leads to a trivial steady state $\rho_{\mathrm{SS}} = \mathbb{I}$. The noisy spin model of Ref. [28] is a particular case of the general RG integrable models corresponding to the limit $x_i = 1$, $z_i^a = \delta_{a,1}$ (see Eq. (12) below).

The vector representation of the Lindblad master equation [33] doubles the Hilbert space $\mathcal{H}$ of dimension $\mathcal{N}$ by mapping the density matrix $\rho$ into a vector $|\rho\rangle$ in the space $\mathcal{H} \otimes \mathcal{H}$ of dimension $\mathcal{N}^2$. The mapping transforms every spin operator $S$ acting to the right or to the left of the density matrix into a superoperator of the form

$$\begin{aligned} S\rho &\to S \otimes \mathbb{I} |\rho\rangle \doteq S |\rho\rangle, \\ \rho S &\to \mathbb{I} \otimes S^T |\rho\rangle \doteq J |\rho\rangle. \end{aligned} \tag{4}$$

Thus, the operators acting to the right of the density matrix are mapped to a new set of spin operators $J_i$ acting on a dual space with the same dimension $\mathcal{N}$ of the original space. This mapping allows the Lindblad master equation to be written in a matrix-vector form $\mathcal{L}|\rho\rangle$ with a Liouvillian matrix $\mathcal{L}$ of dimension $\mathcal{N}^2$ whose eigenvalues can be computed by an exact diagonalization for small size systems or by an appropriate many-body approximation [34–37].

The complete Liouvillian in the Lindblad form is written, after a $\pi$ rotation around the $y$ axis in the dual space, as

$$
\begin{aligned}
\mathcal{L} = & -i\sum_i h_i\left(S_i^z + J_i^z\right) - \Gamma_0\sum_{a=1}^{n_j}\sum_{ij} z_i^a z_i^a\left(S_i^z + J_i^z\right)\left(S_j^z + J_j^z\right) \\
& -\Gamma\sum_{ij} x_i x_j \frac{1}{2}\left[\left(S_i^+ + J_i^+\right)\left(S_j^- + J_j^-\right) + h.c.\right].
\end{aligned}
\tag{5}
$$

The fact that the spin $S_i$ and the dual spin $J_i$ appear always as a sum is a consequence of the equality of the gain and loss $\Gamma_+ \equiv \Gamma_-$ jumps operators. We can, therefore, define the total spin operators $K_i = S_i + J_i$ for each lattice site $i$. In terms of these new total spin operators the Liouvillian acquires the simpler expression

$$
\begin{aligned}
\mathcal{L} = & -i\sum_i h_i K_i^z - \Gamma\sum_i x_i^2\left(K_i\right)^2 + \sum_i\left[\Gamma x_i^2 - \Gamma_0\sum_a\left(z_i^a\right)^2\right]\left(K_i^z\right)^2 \\
& -\Gamma_0\sum_{a,i\neq j} z_i^a z_i^a K_i^z K_j^z - \Gamma\sum_{i\neq j} x_i x_j \frac{1}{2}\left(K_i^+ K_j^- + K_i^- K_j^+\right).
\end{aligned}
\tag{6}
$$

We stress that the magnitude of the spins at each site of the chain $s_i$ is completely arbitrary, with the only constraint imposed by the mapping is that it equals the magnitude of the dual spin $s_i = j_i$. Hence, the total spin at site $i$ can take the following values $k_i = 0, 1, \cdots, 2s_i$.

The Liouvillian (6) presents a set of $L + 1$ weak symmetries [32], given by operators that commute with the total Liouvillian $[\mathcal{L}, O] = 0$. These weak symmetries are the total magnetization operator $K^z = \sum_i K_i^z$ and the $L$ SU(2) Casimir operators at each site

$$
K_i^2 = \left(K_i^z\right)^2 + \frac{1}{2}\left(K_i^+ K_i^- + K_i^- K_i^+\right) = k_i\left(k_i + 1\right).
\tag{7}
$$

Since all symmetry operators commute $[K^z, K_i^2] = 0$, the Liouvillian matrix is separated into blocks labelled by the set of $L$ quantum numbers $k_i$ and each one, in turn, divided into sub-blocks labelled by the total $k_z$. Due to the symmetry of the Liouvillian, the occurrence of a singlet total spin $k_i = 0$ at site $i$ effectively removes this site from the chain, since the action of any SU(2) generator on this state annihilates it. Therefore, the state with all singlet spins is the steady state of the Liouvillian with 0 eigenvalue.

For the particular case of our Liouvillian (6) with an inhomogeneous magnetic field along the $z$ axis, the eigenstates at opposite symmetry sectors $k_z$ and $-k_z$ are related by an anti-unitary operator $F = \mathcal{C}e^{i\pi K^x}$, where $\mathcal{C}$ is the complex conjugation operator and $e^{i\pi K^x}$ performs a $\pi$ rotation around the $x$ axis reversing the spins along the $y$ and $z$ directions. The operator $F$ commutes with the Liouvillian and with the $L$ Casimir operators $K_i^2$ but it does not commute with the $K^z$ symmetry. In spite of the fact that $F$ cannot be added to the chain of weak symmetries, it is useful to understand the structure of the spectrum in each $k_z$ block. For every Liouvillian eigenstate $|\rho_i\rangle$ (in vector form) with complex eigenvalue $\lambda_i$, the eigenstate $|F\rho_i\rangle$ is also a Liouvillian eigenstate with conjugate eigenvalue

$$
\mathcal{L}|F\rho_i\rangle = \mathcal{L}F|\rho_i\rangle = F\mathcal{L}|\rho_i\rangle = F\lambda_i|\rho_i\rangle = \lambda_i^*|F\rho_i\rangle.
\tag{8}
$$

Moreover, the $|F\rho_i\rangle$ state has opposite total angular momentum $K^z$ to $|\rho_i\rangle$

$$\langle F\rho_i|K^z|F\rho_i\rangle = \langle\rho_i|F^\dagger K^z F|\rho_i\rangle = \langle\rho_i|(-K^z)|\rho_i\rangle. \tag{9}$$

Thus, each Liouvillian block with momentum $k_z$ contains the complex conjugate eigenvalues of the corresponding block with momentum $-k_z$. Blocks with $k_z = 0$ are exceptional, since they contain all complex conjugate pairs. This could be explained by the fact that in the subspaces of $k_z = 0$ the operators $F$ and $K^z$ do commute.

The Liouvillian (6) can be derived from the integrals of motion (IOM) of the Richardson-Gaudin models [38, 39]

$$R_i = K_i^z - G\sum_{j(\neq i)=1}^{L} \frac{X_{ij}}{2}\left(K_i^+ K_j^- + K_j^+ K_i^-\right) + Z_{ij}K_i^z K_j^z. \tag{10}$$

The local spins $k_i$ are in principle arbitrary. A possible choice for the matrices $X$ and $Z$ such that the $L$ IOM commute among themselves, $[R_i, R_j] = 0$, is [39]

$$
\begin{aligned}
X_{ij} &= \frac{\sqrt{(1-\alpha)+\alpha\,\eta_i^2}\sqrt{(1-\alpha)+\alpha\,\eta_j^2}}{\eta_i - \eta_j}, \\
Z_{ij} &= \frac{(1-\alpha)+\alpha\,\eta_i\eta_j}{\eta_i - \eta_j},
\end{aligned}
\tag{11}
$$

with $\alpha \in [0,1]$ an interpolation variable within the region of integrability. Note that the limits $\alpha = 0, 1$ assure the equality of the two matrices $X$ and $Z$ and belong to the rational or XXX RG family [40]. The IOM with $\alpha = 0$ lead to the constant pairing Hamiltonian originally solved by Richardson [41], while $\alpha = 1$ leads to the recently studied separable pairing Hamiltonian [42]. For intermediate values of $\alpha$ the matrices $X$ and $Z$ are not equal, corresponding to the XXZ or hyperbolic RG family. Moreover, the IOM have a set of $L+1$ parameters $(G, \eta_i)$ that are arbitrary complex numbers. In what follows, we consider the $\eta's$ as real parameters and $G$ as pure imaginary, $G = ig$. While this choice makes the IOM no longer Hermitian, the integrability properties of these models are kept intact. In terms of these IOM we can obtain the set of integrable Liouvillians (6) with the following linear combination

$$
\begin{aligned}
\mathcal{L}_{\text{int}} = -i\sum_{i=1}^{L}\eta_i R_i = {}& -i\sum_i^{L}\eta_i K_i^z - \frac{g}{2}\sum_{i\neq j}^{L}\left[(1-\alpha)+\alpha\,\eta_i\eta_j\right]K_i^z K_j^z \\
& -\frac{g}{2}\sum_{i\neq j}^{L}x_i x_j \frac{1}{2}\left(K_i^+ K_j^- + K_i^- K_j^+\right),
\end{aligned}
\tag{12}
$$

where $x_i = \sqrt{(1-\alpha)+\alpha\,\eta_i^2}$.

This family of integrable Liouvillians can be derived from am open quantum system with Hamiltonian $H = \sum_i \eta_i S_i^z$, gain and loss jumps (3)

$$L_\pm = \sqrt{\frac{g}{2}}\sum_{i=1}^{L}\sqrt{(1-\alpha)+\alpha\,\eta_i^2}\,S_i^\pm, \tag{13}$$

and two dephasing jumps

$$L_z^1 = \sqrt{(1-\alpha)\frac{g}{2}}\sum_{i=1}^{L}S_i^z, \quad L_z^2 = \sqrt{\alpha\frac{g}{2}}\sum_{i=1}^{L}\eta_i S_i^z. \tag{14}$$

In order to break integrability we will, in addition, consider a set of $n_j$ random jumps that lead to a chaotic (non-integrable) Liouvillian. These jumps are given by

$$L_\pm^a = \sqrt{\Gamma} \sum_{i=1}^{L} w_i^a S_i^\pm, \quad a = 1, \dots, n_j, \tag{15}$$

with $(w_1^a, w_2^a, \dots)$ a set of random orthonormal vectors that are also orthonormal to the integrable gain and loss jump vector $(x_1, x_2, \dots)$. Our results show that this choice of orthonormal $w^a$ vectors displays a more definite chaotic behavior than non orthogonal ones. The chaotic Liouvillian is then

$$\begin{aligned}
\mathcal{L}_{\text{chaotic}} = & -i \sum_i \eta_i K_i^z + \Gamma \sum_{a=1}^{n_j} \sum_i \left(w_i^a\right)^2 \left[\left(K_i^z\right)^2 - (K_i)^2\right] \\
& - \Gamma \sum_{a=1}^{n_j} \sum_{i \neq j} w_i^a w_j^a \frac{1}{2} \left(K_i^+ K_j^- + K_i^- K_j^+\right).
\end{aligned} \tag{16}$$

When choosing the number of random orthogonal jumps one has to take into account that when $n_j$ is equal to the total number of spins $L$, the full Liouvillian becomes diagonal and, thus, integrable. So, the most chaotic spectral statistics occurs for $n_j \approx L/2$ and this has been our choice in Eq. 16.

The transition from integrability to chaos will be characterized by two parameters $(\alpha, \beta)$, that can interpolate between the two rational integrable liouvillians and the fully chaotic limit.

$$\mathcal{L}(\alpha, \beta) = (1 - \beta) \mathcal{L}_{\text{int}}(\alpha) + \beta \mathcal{L}_{\text{chaotic}}, \tag{17}$$

with $(\alpha, \beta) = (0, 0)$ the rational constant model, $(1, 0)$ the separable model and $(\alpha, 1)$ the chaotic limit.

## 3 Liouvillians Spectral statistics in the transition from integrability to chaos

In this work we study the spectrum of Liouvillians of spin $1/2$ chains. Because the $L + 1$ weak symmetries block diagonalize the Liouvillian space, we will focus on the $k_i = 1, k_z = 1$ subspaces, as this represents the block with maximal dimension, adequate for the study of spectral statistics. While the subspace $k_z = 0$ is bigger, in principle, than $k_z = 1$, its spectrum is divided in half by the $F$ symmetry and results into two blocks of smaller dimension. To preserve the scaling of the Liouvillian spectral characteristics with the system size we fix $\Gamma = \frac{1}{L}$ in the subsequent computations.

Figs. 1(a,b) show some typical examples of the spectrum of the rational constant and separable models, respectively, for Liouvillians with $L = 6$, while Fig. 1(c) shows the spectrum of the chaotic Liouvillian limit with the same parameters and $n_j = L/2 = 3$ random jumps. The parameters $\eta_i$ are chosen as $\eta_i = 1/2 + i/L + \xi_i$, where the values of $\xi_i$ are random variables coming from an uniform distribution $\xi_i \in [-1/L, 1/L]$. This is a convenient choice for random values of $\eta_i$ that approximately keeps the relative strength of the Hamiltonian and the jumps constant as we increase size. For the chaotic jumps, the values $\omega_i^a$ are drawn from a uniform distribution $[-1/2, 1/2]$ and then orthonormalized using a Grand-Schmidt procedure. The ordered patterns of the two integrable spectra, as we will see below, have random local distribution of the eigenvalues, while the chaotic spectrum presents level repulsion.

To characterize the integrable to chaos Liouvillian transition we measure the nearest neighbor level statistics. Before that, an unfolding procedure is done in order to rescale the local

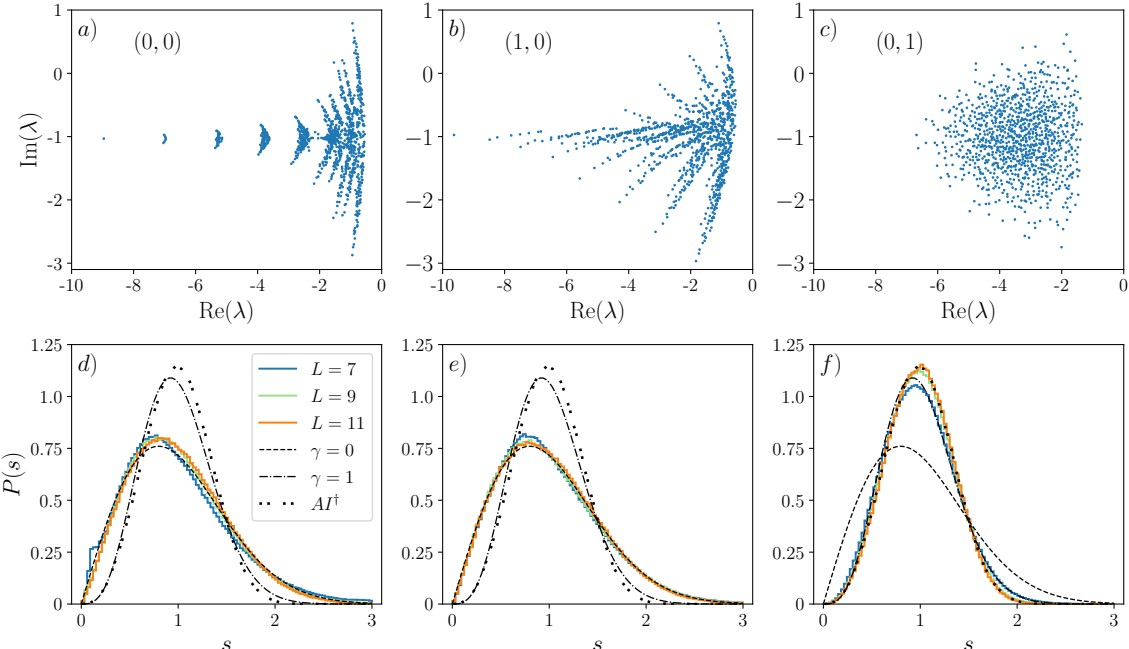

Figure 1: (a-c) Integrable and chaotic Liouvillian spectra for $L = 6$, $K^z = 1$, $k_i = 1$ in the rational constant (a), rational separable (b) and chaotic (c) Liouvillian limits. (d-f) Level spacing distribution $P(s)$ of the integrable and chaotic Liouvillians for different sizes $L = 7, 9, 11$ and $k_i = 1, k_z = 1$ with the same ordering as in (a-c). The dashed and dash-dotted lines show the $\gamma = 0$ and $\gamma = 1$ limits of the interpolation distribution $P(s; \gamma)$ (19), respectively, while the dotted line shows the level statistics of the $AI^\dagger$ universality class.

level density to unity, which results in adimensional quantities to be able to compare different Liouvillian systems to the universal results of Random Matrix Theory [6]. The unfolding procedure can be tricky and if it is not performed carefully may give rise to misleading results [43]. We use a local unfolding procedure where the level distances are rescaled by the local level density [21]. This local density is computed by calculating the area of a circle containing the $n$ nearest neighbors of each level. The nearest-neighbor spacings are then

$$S_i = \sqrt{\frac{n}{\pi d_{i,n}^2}} d_{i,1},$$ (18)

where $d_{i,n}$ is the distance in the complex plane between level $i$ and its $n$-th neighbor. Then, the value of $S_i$ is rescaled by its mean to obtain the final unfolded spacings $s_i$, $s_i = S_i / \langle S_i \rangle$. The unfolded spacings, by construction, have $\langle s \rangle = 1$ as required for a proper comparison. The distribution of $s_i$, $P(s)$, is then contrasted with the theoretical results as explained in the paragraph below. There is a certain arbitrariness in choosing the number $n$, but it has to be large enough to not depend on small scale fluctuations and small enough to capture the local statistics. We have found $n = 20$ to be a good compromise between these two limits, the results being quite stable and robust in a wide range around this value.

For integrable Liouvillians the level statistics follow a Poisson distribution in the plane $P(s) = \frac{\pi}{2} s \exp\left(-\frac{\pi}{4} s^2\right)$, while the level statistics of chaotic Liouvillians belong to the $AI^\dagger$ universality class of non-Hermitian symmetric matrices [24, 25]. To observe the transition from quantum integrability to chaos we fit the level spacing distribution $P(s)$ to a distribution gov-

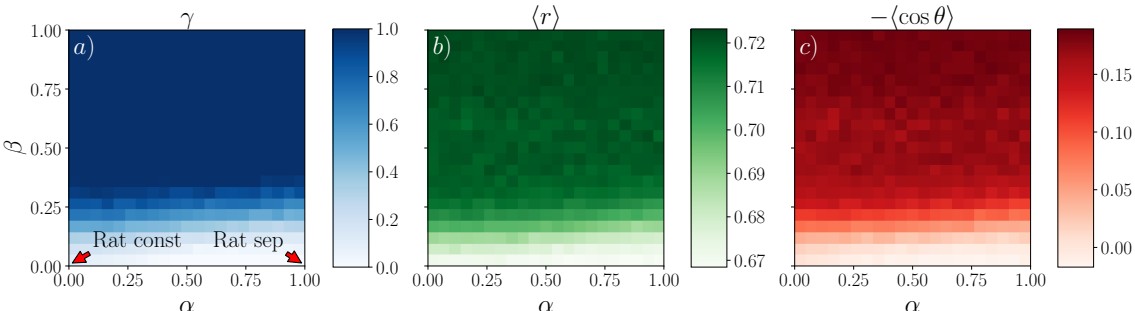

Figure 2: Characterizations of the integrable to chaotic transition using the $\gamma$ [19] parameter (a) and the $\langle r \rangle$ (b) and $-\langle \cos\theta \rangle$ (c) averages of the complex spacing ratios [20]. Each point represents a sample of 20 Liouvillians of matrix dimension 8350.

erned by a parameter $\gamma$

$$P(s;\gamma) = A(\gamma)s^{2\gamma+1}e^{-B(\gamma)s^2}, \quad \gamma \in [0,1], \tag{19}$$

with $A(\gamma), B(\gamma)$ such that $\int_0^\infty P(s)ds = \int_0^\infty sP(s)ds = 1$. Similar to the Brody distribution for the transition between integrability and chaos in standard Hermitian Hamiltonians [18], the limits of this distribution go from the integrable limit $\gamma = 0$, which corresponds to the Poisson distribution in the plane, to the chaotic limit $\gamma = 1$, corresponding to the Wigner surmise of the level spacing distribution of the Ginibre ensemble [23]. Figs. 1(d-f) shows the limit $\gamma = 0$ (dashed line) and $\gamma = 1$ (dash-dotted). While the distribution in the limit $\gamma = 1$ does not exactly correspond to the level statistics of the $AI^\dagger$ universality class (dotted line in Fig. 1), it shows the same level repulsion $P(s) \sim s^3$ and both distributions are sufficiently similar so that $\gamma = 1$ is a good characterization of the chaotic limit.

The unfolded level statistics of the two rational and chaotic limits are shown in Figs. 1(d-f) for Liouvillians with sizes $L = 7, 9, 11$ and Hilbert space dimension 357, 2907 and 24068, respectively. For each of these sizes $L$ we compute a sample of several Liouvillian spectra so that there are in total more than 700.000 eigenvalues to draw statistics from. The two integrable limits show levels statistics that are very close to the 2d Poisson distribution in the plane, indicating a random distribution of the local eigenvalues, while the chaotic limit shows similar level statistics to the $AI^\dagger$ universality class. For each Liouvillian limit there is also a convergence to the respective 2d Poisson and $AI^\dagger$ distributions from smaller to larger $L$'s.

Fig. 2(a) shows the transition from integrable to chaotic level statistics characterized by the interpolation parameter $\gamma$, which we fit to the level spacings using a Maximum Likelihood Estimation. Each value of the parameters $(\alpha, \beta)$ represents the statistics of a sample of 20 Liouvillians of size $L = 10$, with a Hilbert space dimension of 8350 eigenvalues, and $n_j = 5$ random jumps. For values of $\beta = 0$ the transition parameter $\gamma$ is very close to 0, showing integrable statistics for both rational integrable Liouvillians, and for $\beta > 0$ there is a smooth transition in terms of $\gamma$ from integrable to chaotic statistics up to the limit $\beta = 1$, with the integrability breaking at relatively low values of $\beta$. This transition is shown in more detail in Fig. 3 for different system sizes $L$ and for $\alpha = 0, 0.5, 1$. We have rescaled the transition parameter $\beta$ by $L^2$. Our computations suggest that the fully chaotic limit $\gamma = 1$ is reached at a critical point $\beta_c(\alpha) = C(\alpha)L^{-2}$, with $C(\alpha)$ a constant that only depends on the properties of the integrable Liouvillian [12]. This points to a sharp transition from integrability to chaos in the thermodynamical limit as soon as random Lindblad jumps [15] are added to the system at $\beta > 0$. The transition occurs earlier for $\alpha = 0$ probably because there is a wider distribution of

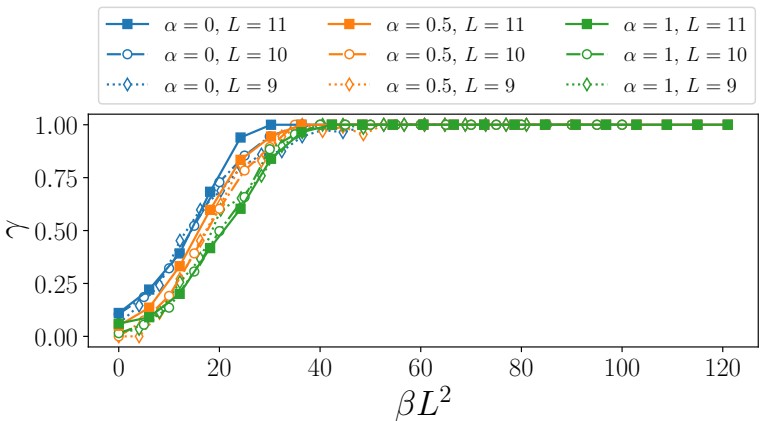

Figure 3: Dependence of the fitting parameter $\gamma$ on the transition parameter $\beta$ rescaled with $L^2$ for the three Liouvillian limits: rational constant $\alpha = 0$, general hyperbolic $\alpha = 0.5$ and separable $\alpha = 1$, and for different system sizes $L = 9, 10, 11$.

Hamiltonian parameters in the integrable part of the model with $\alpha = 1$. However, the system sizes we are able to reach are not large enough to make a definite claim in this regard.

Apart from the interpolation distribution $P(s; \gamma)$ we study the integrable to chaos transition using the complex spacing ratios recently introduced by Sá *et al.* [44]

$$z_k = \frac{\lambda_k^{\text{NN}} - \lambda_k}{\lambda_k^{\text{NNN}} - \lambda_k},$$ (20)

with $\lambda_k$ the $k$-th eigenvalue and $\lambda_k^{\text{NN}}, \lambda_k^{\text{NNN}}$ its first and second neighbors, respectively. The advantage of the spacing ratios is that, because they are adimensional quantities, the unfolding procedure is not needed, at least if the local density of states does not vary in the scale of the typical minimal distance between levels. The equivalent quantity for closed systems introduced in Ref. [45] has become widely used for this reason. The complex spacing ratios have only been computed for a few examples [31, 44, 46] yet so it is particularly important to compare their results to the more standard spacing distributions in the integrable to chaotic transition in Liouvillians.

Figs. 2(b,c) show the means of the absolute and phase angle values, respectively, of the complex ratios $z_k = r_k e^{\theta_k}$, which were determined by Sá *et al.* to be good indicators of the transition. They find in the limit $L \to \infty$, the values $(\langle r \rangle, -\langle \cos \theta \rangle) = (2/3, 0)$ for integrable statistics and $\approx (0.72, 0.2)$ for the $AI^\dagger$ universality class. Our results for $\langle r \rangle$ [Fig. 2(b)] coincide with these limits for both integrable $\beta = 0$ and chaotic $\beta = 1$ families of Liouvillians, while $-\langle \cos \theta \rangle$ has a slow convergence to the infinite size limit, also mentioned by Sá *et al.* and its limits are a bit lower than those of $L \to \infty$. Apart from the finite size effects, both measurements of the complex spacing rations show a good agreement with the $P(s; \gamma)$ interpolation distribution.

## 4 Conclusions

To summarize, we have presented a new continuous family of integrable many-body Liouvillians based on the rational and hyperbolic Richardson-Gaudin models. In vector form, these Liouvillians can be written as a fully coupled network of spins of the XXX or XXZ type, representing open quantum spin systems with gain, loss, and dephasing jumps. This very general family of integrable Liouvillians increases greatly the number of systems that can be inves-

tigated with exact solutions, which can open new avenues of research for the study of the properties of open quantum many-body systems, a topic that is still in its infancy.

By adding chaotic jumps, we define two-parameter Liouvillians that describe the transition between the different integrable models and chaotic non-integrable Liouvillians with the same degrees of freedom. We have characterized these transitions studying the spectral statistics of the Liouvillians's eigenvalues in the complex plane using two methods, the nearest-neighbor spacing distribution and the average properties of the complex spacing ratios. The complex ratios do not require the difficult unfolding procedure, which is an important advantage as in the case of complex spectra the unfolding is even more critical than for real spectra.

Within the integrable line, we find excellent agreement between the spacing distribution and the distribution of a Poisson process in the plane, while in the chaotic case we find a very good agreement with the $AI^\dagger$ random matrix universality class. The agreement improves as we reach larger sizes. We have derived a simple interpolation formula between the linear and cubic repulsion cases as a function of one parameter. The formula captures nicely the transition between integrable and chaotic dynamics in the spacing distribution of the complex Liouvillian eigenvalues, allowing a quantitative description. Our results show that the Liouvillian transition to chaotic dynamics with the addition of random Lindblad jumps occurs faster as the size of the system is increased, pointing to a sharp transition in the thermodynamic limit as soon as the random jumps are added, although we cannot reach large enough sizes to make a definite commitment. In the case of the ratios, the behavior of the average value of their modulus and phase shows excellent agreement with the behavior of the fitted parameter value of the interpolation formula as we scan the transition between integrability and chaos in the parameter space of our model. These results give an extra support for the use of the complex ratios for characterizing the onset of chaos in open quantum systems. A similar analysis in other models could help to elucidate the role of chaos in information scrambling and thermalization in quantum many body open systems. We anticipate that the applications of these tools will be as widespread as the equivalent spectral analysis in closed quantum many body systems.

# Acknowledgements

**Funding information** We acknowledge financial support from Projects No. PGC2018-094180-B-I00 (MCIU/AEI/FEDER, EU) and and CAM/FEDER Project No. S2018/TCS-4342 (QUITEMAD-CM). This research has been also supported by CSIC Research Platform on Quantum Technologies PTI-001.

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
