# Peer review of "From integrability to chaos in quantum Liouvillians"

_SciPost Physics Core, doi:SciPost Phys. Core 5, 026 (2022)_

## Round 1 · Referee Report · Anonymous (Referee 1) · 2022-2-1

Report

I thank the authors for the new analysis of their numerical data. However, in my opinion, the new version of Fig. 3 brings little added value. In its current form, the collapse of the curves for different system sizes is not evident and might simply be an artifact of the horizontal scale used for the plot. More seriously, the analysis made by the authors shows that there is no transition (with a transition point $\beta\ne 0$) from integrability to chaos in the thermodynamic limit. The smooth transition from Poisson to RMT level statistics is probably simply due to finite size effects. Only at $\beta=0$ does the system display Poissonian level statistics, which simply follows from the fact that the model is integrable by construction at this point. In conclusion, the proposed model does not seem to exhibit the expected transition from integrability to chaos in the thermodynamic limit, and the original goal of this work seems to me to be compromised. I do not see what significant practical or conceptual value the model proposed by the authors could have and therefore, I cannot recommend this work for publication.
  • validity: high
  • significance: low
  • originality: ok
  • clarity: good
  • formatting: reasonable
  • grammar: good

Author:  Rafael Molina  on 2022-02-03  [id 2152]

(in reply to Report 1 on 2022-02-01)
Category:
reply to objection

We thank the referee for the report on our manuscript. We disagree, however, on the analysis of the last results about the scaling shown in Fig. 3. Although, the results show that in the thermodynamic limit the transition to chaos is abrupt at infinitisimal value of beta, the scaling of Fig.3 shows that the transition occurs for any finite system with a proper scaling of the parameters and the transition, then, is relevant for real systems which are always finite. The goal of the article is the study of the transition to chaos in Liovillians and we believe we have achieved that goal. In the process we have shown the usefulness of the spectral statistics tools used in our study and presented new families of integrable Liouvillians. We need to stress than in transition models typically used in closed quantum systems, an abrupt transition to chaos in the thermodynamic limit is also observed. In spite of that, the transition can be observed experimentally and it has been observed in many different systems. Check, for example, the classic work by Lenz et al. Scaling laws of the additive random-matrix model Physical Review A 44, 8043 (1991). Please, also note that the preprint version of our manuscript has already proven valuable to the community as it has already 4 citations according to google scholar or 5 citations according to NASA ADS web page.

---

## Round 1 · Referee Report · Anonymous (Referee 3) · 2022-2-14

Strengths

  • very clearly written paper, provides an excellent motivation and introduction to the concept of integrable RG-like Liouvillians
  • careful analysis and clear presentation of the numerical results

Weaknesses

  • unclear originality

Report

The paper reports on careful implementation of the analysis of spectral correlations for characterization of the integrability to chaos transition in Richardson-Gaudin like Lindblad equations. The paper provides and excellent discussion of integrable RG-like Lindblad equations.

The paper is certainly valuable to the field, but it is hard to identify it as very original, specifically it does not really open any new research avenue nether it defines any new concept or provide a solution to an important problem. Nevertheless, I recommend publication, but perhaps instead in SciPost Physics Core.

Requested changes

  • Can the authors substantiate their claim that the statistics becommes less chaotic by introducting more (than L/2) random jump operators (15)? I find it rather surprising?

  • p4: F is not a weak symmetry, strictly, since it is anti-unitary.

  • p5: the set integrable Liouvillians -> the set of integrable Liouvillians

  • validity: high
  • significance: high
  • originality: ok
  • clarity: top
  • formatting: excellent
  • grammar: excellent

Author:  Alvaro Rubio-García  on 2022-02-18  [id 2221]

(in reply to Report 2 on 2022-02-14)
Category:
answer to question

We thank the Referee for their insightful observations. We agree in that the F operator is not strictly a weak symmetry due to it being anti-unitary. However, due to its usefulness in relating the Liouvillian spectrum's symmetry sectors, we have rewritten the paragraph around Eqs. (8,9) without any mention to weak symmetries. With respect to the number of random jumps, if we were to have L orthonormal jumps, then the Liouvillian would become diagonal and integrable. Thus, we choose the number of random jumps to be of order L/2, for which the most chaotic statistics occur. We have rewritten the paragraph after Eq. (16) accordingly.

We attach to this comment a new version of the manuscript.

Following the Referee's suggestion of the publication of this manuscript in SciPost Physics Core, we would like to ask the Editor for a transfer to that journal.

Attachment:

220216-manuscript_scipost.pdf

---

## Round 1 · Author Response

We thank both referees for the careful reading of the manuscript and for their constructive criticism. We have made the best effort to comply with the requested changes. We believe this new version might be ready for publication. We respond to the specific questions and requests of the referees below.

Anonymous Report 1

Following the advice of the referee (which was similar to a request of the second referee) we have further analyzed the transition from integrability to chaos in more detail. Specifically, we have studied the transition for different values of the parameter alpha as a function of beta and the size of the system. Unfortunately, the matrix dimensions grow too fast (as usual in many-body quantum systems) to be able to study a span of different sizes large enough to make a proper size scaling. We have concentrated on the extreme values of alpha and have added a figure of the increase in gamma, the parameter measuring the transition to chaos, as beta is increased for alpha=0 and alpha=1. We have made smaller calculations for other values of alpha that point to similar results in those cases. The results point to a sharper transition to chaos as the system size is increased. This may indicate that an actual phase transition occurs but the reached system sizes are not large enough to make a definite claim.

Anonymous Report 2

Concerning the weakness and the requested changes we agree that the many-body character of the Liouvillian is the novelty of the work and that this is the main problem for a theoretical understanding as the relationship between the statistical measures and quantum chaos is only well founded in semiclassical theory. This was not stressed enough in the original version of the introduction. We have added several sentences in the first paragraphs of the introduction regarding this issue and we have also added an explanation of the semiclassical theory of Berry and Robnik for the intermediate statistics.

We have also studied in more detail the dependence of the fitting parameter \gamma with the strength \beta of the chaotic term. We have added a new Fig. 3 in the results section, and a new sentence in the conclusions. This point has been also the main request of the first report and a more detailed answer in this regard can be found there.

---

## Round 1 · List of Changes

1- In the introduction, it should be mentioned that P(s)∝sβ is the expected behaviour for small values of s

We have added the sentence "for small s" after the mention of "P(s)∝sβ" in the first paragraph.

2- in the second column of the Introduction, “… an important important step …” should read “… an important step …”

This typo has been corrected in the new manuscript version.

3- the discussion following Eq. (1) on the Liouvillian spectrum and the steady state (which may not be unique) is not accurate when the zero Liouvillian eigenvalue is degenerate

We have added a sentence at the end of the first paragraph of Section 2 mentioning the possibility of having degenerate steady states and what happens in this case.

4- In Eq. (3), the numbers zai and xi are not defined in the text

These parameters are now defined after Eq. (3).

5- In the last line of the paragraph following Eq. (6), "... at site ki can ..." should read "... at site i can ..."

This error is now corrected in the new version of the manuscript.

6- In Eq. (16), my impression is that the first term of the right-hand side should correspond to Eq. (12), which is not the case in the manuscript

While both equations by themselves are right, it would be clearer for the reader if the first terms of Eqs. (12) and (16) are the same. We have modified the first term of Eq. (16) so that they now coincide.

7- The authors should justify the choice of orthonormal vectors for the factors entering the chaotic and integrable parts [as described after Eq. (15)]

This choice has been made in light of our numerical results using both orthonormal and non orthonormal vectors. We observed that orthonormal vectors showed stronger signatures of chaos than non orthonormal ones. A clarifying sentence has been added in the paragraph after Eq. (15).

8- In Eq. (17), Lint(α) is not defined, although it should correspond to Eq. (12)

We now define L_int(\alpha) at the left hand side of Eq. (12).

9- In Sec. III, the text describing the unfolding procedure of the spectrum is too short and vague to be clear

We have expanded the description of the unfolding procedure around Eq. (18). We think it now gives a clear description of the unfolding procedure used in this work.

10- The caption of Fig. 1 does not provide sufficient details. How are the random values of the parameters wi in Eq. (15) drawn ? Is there a link with the ωi used for the ηi parameters ?

We have added a sentence in the second paragraph of Section 3 detailing that the ωi parameters are drawn from a uniform distribution in the range [-1/2, 1/2] and that we then orthonormalize the vectors using a Grand-Schmidt procedure. The ωi parameters are not related to the ηi ones and we have changed the definition of the ηi parameters so that there is no room for confusion.

11- In Fig. 2, colorbars for the γ, ⟨r⟩ and −⟨cosθ⟩ values are missing

The colorbars have been added to the Figure.

12 -

Following the comments made by both Referees we have added a new Fig. 3 in the results section with the growth of the parameter \gamma as the transition parameter \beta is increased. We have also added a new sentence in the last paragraph of the conclusions commenting these results.

---

## Round 2 · List of Changes

• In the discussion of the F operator above Eq. (8) we now explicitly mention it is an anti-unitary operator and refer to it as an operator instead of as a weak symmetry.

  • In the new version of the manuscript we now explain below Eq. (16) the reasons that lead us to choose n_j ~ L/2, mainly that introducing as many (orthonormal) random jumps as number of spins leads again to a diagonal - and thus integrable - Liouvillian.

  • We fixed above Eq. (12): "the set integrable Liouvillians" -> "the set of integrable Liouvillians"

---

## Editorial Decision

published